# Design, Synthesis and Biological Evaluation of *Syn* and *Anti*-like Double Warhead Quinolinones Bearing Dihydroxy Naphthalene Moiety as Epidermal Growth Factor Receptor Inhibitors with Potential Apoptotic Antiproliferative Action

**DOI:** 10.3390/molecules27248765

**Published:** 2022-12-10

**Authors:** Essmat M. El-Sheref, Mohamed A. Ameen, Kamal M. El-Shaieb, Fathy F. Abdel-Latif, Asmaa I. Abdel-naser, Alan B. Brown, Stefan Bräse, Hazem M. Fathy, Iqrar Ahmad, Harun Patel, Hesham A. M. Gomaa, Bahaa G. M. Youssif, Asmaa H. Mohamed

**Affiliations:** 1Chemistry Department, Faculty of Science, Minia University, El Minia 61519, Egypt; 2Chemistry Department, Florida Institute of Technology, 150 W University Blvd, Melbourne, FL 32901, USA; 3Institute of Biological and Chemical Systems, IBCS-FMS, Karlsruhe Institute of Technology, 76131 Karlsruhe, Germany; 4Pharmaceutical Organic Chemistry Department, Faculty of Pharmacy, Al-Azhar University, Assiut Branch, Assiut 71524, Egypt; 5Division of Computer Aided Drug Design, Department of Pharmaceutical Chemistry, R. C. Patel Institute of Pharmaceutical Education and Research, Shirpur 425405, Maharashtra, India; 6Department of Pharmaceutical Chemistry, Prof. Ravindra Nikam College of Pharmacy, Gondur, Dhule 424002, Maharashtra, India; 7Pharmacology Department, College of Pharmacy, Jouf University, Sakaka 72314, Saudi Arabia; 8Pharmaceutical Organic Chemistry Department, Faculty of Pharmacy, Assiut University, Assiut 71526, Egypt

**Keywords:** azide, naphthalene, click, quinolin-2-one, apoptosis, caspases, antiproliferative, reaction mechanism

## Abstract

Our investigation includes the synthesis of new naphthalene-bis-triazole-bis-quinolin-2(1H)-ones **4a**–**e** and **7a**–**e** via Cu-catalyzed [3 + 2] cycloadditions of 4-azidoquinolin-2(1*H*)-ones **3a**–**e** with 1,5-/or 1,8-bis(prop-2-yn-1-yloxy)naphthalene (**2**) or (**6**). All structures of the obtained products have been confirmed with different spectroscopic analyses. Additionally, a mild and versatile method based on copper-catalyzed [3 + 2] cycloaddition (Meldal–Sharpless reaction) was developed to tether quinolinones to O-atoms of 1,5- or 1,8-dinaphthols. The triazolo linkers could be considered as anti and syn products, which are interesting precursors for functionalized epidermal growth factor receptor (EGFR) inhibitors with potential apoptotic antiproliferative action. The antiproliferative activities of the **4a**–**e** and **7a**–**e** were evaluated. Compounds **4a**–**e** and **7a**–**e** demonstrated strong antiproliferative activity against the four tested cancer cell lines, with mean GI_50_ ranging from 34 nM to 134 nM compared to the reference erlotinib, which had a GI_50_ of 33 nM. The most potent derivatives as antiproliferative agents, compounds **4a**, **4b**, and **7d**, were investigated for their efficacy as EGFR inhibitors, with IC_50_ values ranging from 64 nM to 97 nM. Compounds **4a**, **4b**, and **7d** demonstrated potent apoptotic effects via their effects on caspases 3, 8, 9, Cytochrome C, Bax, and Bcl2. Finally, docking studies show the relevance of the free amino group of the quinoline moiety for antiproliferative action via hydrogen bond formation with essential amino acids.

## 1. Introduction

Over the past few decades, quinolones have transformed from a small and insignificant class of drugs primarily utilized for treating mild urinary tract infections to some of the most prescribed antibacterials globally [1,2,3,4]. An important different activity for quinolones has been investigated despite being well known as antibacterial. In the late 1980s, quinolone derivatives held significant potency against eukaryotic Type II topoisomerases (topoisomerase II) and demonstrated cytotoxic activity against cancer cell lines. Hence, quinolinone derivatives are promising candidates for cancer treatment [5,6,7]. Several quinolone derivatives exemplified by voreloxin, AT-3639, and quarfloxin have already been used in clinics or in clinical trials [8,9]. 

In addition, quinolinone is an intriguing fused heterocyclic scaffold that is found in several FDA-approved and commercialized anticancer medications [10], including Neratinib (Nerlynx^®^), an EGFR-TK inhibitor [11] (Figure 1). Moreover, several quinolinone scaffold-containing compounds are potent EGFR inhibitors. Compound **I** has an EGFR IC_50_ of 0.0075 µM [12], compound **II** has an EGFR IC_50_ of 5 nM [13], and compound **III** has an EGFR IC_50_ of 5.2 µM [14,15] (Figure 1). 

Currently, 1,2,3-triazoles exhibit a diverse set of biological actions [16,17,18,19]. They possess various pharmacological and biological features, including anticancer action [20]. Additionally, the importance and applications of 1,2,3-triazole compounds have increased [21,22], after development of the reactions of organic azides with terminal alkynes under mild conditions catalyzed by Cu(I). Further, the regioselective formation of 1,2,3-triazoles using Cu-catalyzed [3 + 2] cycloaddition has proved to be the best example of click chemistry with extensive applications in organic and medicinal chemistry [23]. However, several investigations have shown that their biological features are attributable to the triazole moiety’s making of hydrogen bonds, dipole–dipole, and stacking interactions, which warrant the development of stable complexes and, as a result, activate a cascade of metabolic activations such as apoptosis [24,25]. As a result, the antiproliferative activity of 1,2,3-triazole derivatives is explained by different mechanisms of action. Perihan and coworkers, for example, used a multi-target design technique to develop various 1,2,3-triazoles, one of which, compound **IV** (Figure 2), has been demonstrated to arrest the G2/M cell cycle and induce apoptosis in human cancer cells [26]. In addition, Khan and colleagues [27] described novel diphenyl-1*H*-pyrazole-based acrylates linked to 1,2,3-triazole **V** (Figure 2) as prospective apoptosis-inducing cytotoxic agents.

We recently reported on compound **VI**’s design, synthesis, and antiproliferative activity (Figure 2) [28]. Compound **VI** displayed a substantial antiproliferative activity against four cancer cell lines, with a mean GI_50_ = 0.23 µM. Compound **VI** inhibited EGFR activity with an IC_50_ of 0.11 µM. The apoptotic mechanism demonstrated that compound **VI** increased Caspase-3, Caspase-9, and Cytochrome-C levels in human (Panc-1) cancer cells by 7.80, 19.30, and 13 times, respectively, compared to doxorubicin. Furthermore, **VI** elevated Bax levels to 40-fold greater than normal untreated cells while decreasing anti-apoptotic Bcl-2 levels 6.3-fold.

Hybridization has emerged as a promising strategy in developing new drugs with the potential to overcome cross-resistance and improve affinity and efficacy compared with the parent drugs [29,30]. Combining quinolinone with other anticancer pharmacophores may provide new candidates with great potency against drug-sensitive and drug-resistant cancers. 

Motivated by the facts presented here, we present the synthesis of a small set of quinoline-1,2,3-triazole hybrids that will be evaluated for antiproliferative activity. The newly synthesized compounds consist of two scaffolds: **Scaffold A** (**4a**–**e**), which represents *Syn*-like-quinoline derivatives, and **Scaffold B**, which represents *Anti*-like-quinoline derivatives **7a**–**e**, Figure 3. The two ligands of triazole and quinolinone moieties attached to the naphthalene core of Syn and Anti conformation-like represented diversity in the field of biology. 

The antiproliferative activity of compounds **4a**–**e** and **7a**–**e** will be investigated using four cancer cell lines: A549 (epithelial cancer cell line), MCF-7 (breast cancer cell line), Panc-1 (pancreas cancer cell line), and HT-29 (colon cancer cell line). The most effective antiproliferative agents will be examined further for their potential inhibitory activity against EGFR as a target for their mechanistic action. Furthermore, the most potent derivatives will be tested for their ability to trigger apoptosis against caspases 3, 8, and 9, cytochrome c, Bax, and the anti-apoptotic Bcl2.

## 2. Results and Discussion

### 2.1. Chemistry

Figure 1 and Figure 2 depict the synthesis of compounds **4a**–**e** and **7a**–**e** via the cycloaddition reaction of the obtained 1,8-bis(prop-2-yn-1-yloxy)naphthalene (2) [31], or 1,5-bis(prop-2-yn-1-yloxy)naphthalene (6) with different readily prepared azides of quinolinones **3a**–**e** [32,33,34,35,36]. The cycloaddition reaction of azide–alkyne was performed in DMF via click conditions to afford a series of five different *Syn*-like quinolinone-dioxo-naphthalene hybrids via triazole linker 4**a**–*e*. The characterization data, including ^1^H NMR, ^13^C NMR, ^1^H-^1^H COSY, HMBC, HSQC, and ^15^N NMR spectroscopy, mass spectrometry, and elemental analysis, confirmed the chemical structures of our new synthesized compounds 4**a**–**e**.

To confirm our obtained products, NMR, mass spectrometry, and elemental analysis were performed for all obtained products. All spectral data for synthesized products **4a**–**e**, clarify that they are formed from two molecules of compounds **1a** to **e** and one molecule of terminal alkyne **2** in a molar ratio (2:1) without any elimination. To rationalize our results, we choose compound **4c** which assigned as [4,4′-(((naphthalene-1,8-diylbis(oxy))bis(methylene))-bis(1*H*-1,2,3-triazole-4,1-diyl))bis(6-methoxyquinolin-2(1*H*)-one)] (Figure 4).

Elemental analysis and mass spectrometry agreed with its general formula C_36_H_28_N_8_O_6_ and molecular weight (*m*/*z* = 668). The ^1^H NMR spectrum for compound **4c** showed six-singlet signals at δ_H_ = 12.20 (2H), 9.03 (2H), 6.94 (2H), 6.90 (2H), 5.51 (4H), and 3.67 (6H) ppm., which are assigned as quinolinone-NH, H-5, H-10, H-8, H-4a (-O-CH_2_), and H-11a (OCH_3_) groups, respectively. Further, the methoxy protons H-11a are distinctive at δ_H_ 3.67 ppm; HMBC and HSQC correlation with the carbon appears at δ_C_ 55.33 ppm, which is assigned as C-11a, and other HMBC correlations with the non-protonated carbon appear at δ_C_ 154.51 ppm, which is assigned as C-11. The protons H-4a and H-5 also give HMBC correlation with the nitrogen at δ_N_ 247.6, assigned as N-1. The other sp3 nitrogen, at δ_N_ 151.3, gives only HSQC correlation with H-6; this nitrogen is assigned as N-6. Additionally, the carbon at δ_C_ 126.03 ppm gives three HMBC correlations with the protons appearing at δ_H_ 7.85 (d, *J* = 8.2 Hz; 2H), 7.45 (d, *J* = 8.2 Hz; 2H), and 7.30 (d, *J* = 7.4 Hz; 2H), ppm.; this carbon is assigned as C-4a’ and the signals were assigned as H-4′, H-3′ and H-2′, respectively. Furthermore, the carbon appeared at δ_C_ 125.47 ppm, giving one HSQC correlation with the proton at δ_H_ 7.45 and another HMBC correlation with the protons appearing at δ_H_ 7.85 and 7.30 ppm., and was assigned as C-3′.

On the other hand, the same synthetic methodology was used to produce a new series of bis-quinolinone-1,2,3-triazole hybrids as more analogs for biological testing. The cycloaddition reaction of 4-azido-2-quinolinones **3a**–**e** with highly symmetrical di-*o*-propargylated compound **6**, obtained by propargylation of 1,5-dinaphthol (**5**), was promoted under an above similar condition, resulting in 76–84% yield of the corresponding symmetrical bis-quinolinone triazole-based anti-like targets **7a**–**e** (Figure 2). The core architecture of these bis-quinolinone-triazoles and their diversity tempted us to study the anti-cancer properties of such compounds. 

The structures of 1,5-bis(prop-2-yn-1-yloxy)naphthalene (**6**) were confirmed using different spectral data such as ^1^H NMR, ^13^C NMR, 2D-NMR spectrum, as well as mass spectrometry. For example, it exhibited a molecular formula of C_16_H_12_O_2_, which was compatible with its *m*/*z* = 236. Through the different data, we found that compound **6** fits perfectly with the previous compound **2** with a slight difference in the chemical shift’s values in its ^1^H NMR spectrum, and it also contains eight lines in this ^13^C NMR spectrum, while the previous compound **2** contains nine lines in this ^13^C NMR spectrum (Figure 5). In the ^1^H NMR spectrum of compound **6**, three doublet signals appeared at δ_H_ 7.75, 7.10, and 5.01 ppm, assigned as H-4, H-2, and –OCH_2_, respectively. The doublet–doublet signal at δ_H_ 7.44 ppm gives HSQC and HMBC correlation with the carbon at δ_C_ 125.43 ppm., which is assigned as C-3, and this proton was assigned as H-3. This compound contains only one carbon signal at δ_C_ 125.97 ppm, which gives HMBC correlation with both signals at δ_H_ 7.75, 7.44, and 7.10 ppm., so it must be C-4a. Further, the ^13^C NMR spectrum showed other signals at δ_C_ 152.55, 106.74, 79.12, 78.38, and 55.95 ppm, which were assigned as C-1, C-2, C-1b, C-1c, and –O-CH_2_, respectively. 

On the other hand, the novel synthesizing of naphthalene-bis-triazole-bis-quinolin-2(1*H*)-ones **7a**–**e** was also confirmed on the bases of different spectral data. For example, we choose compound **7e** which was assigned as [4,4′-(((naphthalene-1,5-diylbis(oxy))bis(methylene))bis-(1*H*-1,2,3-triazole-4,1-diyl))bis(1-methylquinolin-2(1*H*)-one)] (Figure 6).

To confirm our results, we choose compound **7e** which was assigned as [4,4′-(((naphthalene-1,5-diylbis(oxy))bis(methylene))bis(1*H*-1,2,3-triazole-4,1-diyl))-bis(1-methylquinolin-2(1*H*)-one)]. Elemental analysis and all spectral data were acceptable with this proposed structure with chemical formula C_36_H_28_N_8_O_4_ (*m/z* = 636). Through the interpretation of the various analyses that were conducted for compound **7e** and all the previous compounds, it is undoubtedly clear that the proposed chemical composition is correct, and also, to prove the composition in another way, it was compared with spectral data for its analogous compound **4e** (Table 1). This is claimed to be the reaction product of 1,8-bis(propargyloxy)naphthalene (**2**) with 4-azido-1-methyl-1*H*-quinolin-2-one (**3e**). Additionally, the spectra are consistent with structure **7e** but are very similar to those observed for sample **4e**, for which **4e** was assigned. The spectroscopic difference between these products is that, in **4e**, C-4a’ and C-8a’ are nonequivalent, while in **7e**, these two carbons are equivalent. However, in both compounds **4e** and **2**, C-8a’ was assigned as co-resonant with another carbon which could not be observed separately. Therefore, we suspect that the correct structure for both products is **4e**; see Table 1. In addition, all spectroscopic analyses of all quinolin-2-one rings are completely identical to what was previously published [37,38,39].

The mechanism for the obtained products **4a**–**e** and **7a**–**e** can be rationalized as, upon mixing (1 mmol) of terminal alkynes **2** and/or **6** with Cu(1), the salt of compound **8** would form. On the addition of (2 mmol) of azides **3a**–**e** dissolved in DMF, a nucleophilic addition takes place to give the adduct **9**, which undergoes nucleophilic attack to the N-negative charged on triple bond to give the adduct **10**, which then further accepts H^+^ and forms the intermediate **11**, which then reacts with Cu(1) to give the final products **4a**–**e** and **7a**–**e** via further repeating the above steps as shown in Figure 3 [23,40].

### 2.2. Biology

#### 2.2.1. Antiproliferative Action

##### Cell Viability Assay

A cell viability test was performed using MCF-10A (human mammary gland epithelial) cell line [41,42,43] to investigate the effect of **4a**–**e** and **7a**–**e** on normal cell lines. In this investigation, a concentration of 50 µM of the studied compound is employed for four days, after which cell viability is assessed. The results showed that compounds **4a**–**e** and **7a**–**e** have no toxic effect and have more than 86% cell viability, as shown in Table 2.

##### Antiproliferative Assay

The newly synthesized compounds were tested for antiproliferative activity against four different types of cancer cells [44,45]: A-549 (epithelial cancer cell line), MCF-7 (breast cancer cell line), Panc-1 (pancreas cancer cell line), and HT-29 (colon cancer cell line). Erlotinib was used as the reference, and Table 2 displays the results of calculating the IC_50_ of each compound.

Generally, the newly evaluated compounds **4a**–**e** and **7a**–**e** displayed significant antiproliferative activity, with mean GI_50_ ranging from 34 nM to 134 nM compared to the reference erlotinib with a GI_50_ of 33 nM against the four tested cancer cell lines. Three compounds with the highest antiproliferative activity were identified: **4a** and **4b** with a Syn quinoline backbone structure (Scaffold A), and **7d** with an Anti-quinoline backbone structure (Scaffold B), with GI_50_ values ranging from 34 nM to 54 nM.

Compound **4a (**R_1_ = R_2_ = R_3_ = H**)** was the most potent synthetic derivative, with a mean GI_50_ of 34 nM compared to the reference erlotinib’s GI_50_ of 33 nM. Compound **4a** inhibited the MCF-7 (breast cancer) cell line more effectively than erlotinib, with an IC_50_ of 33 nM versus 40 nM. The antiproliferative effects of compounds **4e** (R_1_ = CH_3_, R_2_ = R_3_ = H, Syn quinoline backbone) and compound 7e (R_1_ = CH_3_, R_2_ = R_3_ = H, Anti quinoline backbone), which have GI_50_ of 130 and 134 nM and are approximately 4-fold less potent than 4a, were greatly diminished when the free NH group in compound 4a was replaced with a methyl group. This finding highlights the significance of the free amino group at position-1 of the quinoline moiety for the antiproliferative action. 

Compound **4b** (R_1_ = R_2_ = H, R_3_ = CH_3_) ranks second in efficacy against the cancer cell lines tested, with a GI_50_ of 45 nM, which is 1.4-fold less potent than erlotinib (GI_50_ = 33 nM). Compound **7b** (R_1_ = R_2_ = H, R_3_ = CH_3_) shares the same substitution pattern as compound **4b**, but was found to be 1.4 times less potent due to its anti-quinoline backbone structure. This finding emphasizes the impact of stereochemistry in the action of this class of organic compounds, with the syn derivatives being more active than the anti-derivatives. The same pattern becomes apparent when **4a** (GI_50_ = 34 nM) and **7a** (GI_50_ = 130 nM) are compared.

The nature and position of the substitution on the quinoline moiety were also studied. The GI_50_ of the 6-methyl quinoline derivative **4b** (R_1_ = R_2_ = H, R_3_ = CH_3_) was found to be 1.8-fold more potent than that of the 6-methoxy derivative **4c** (R_1_ = R_2_ = H, R_3_ = OCH_3_), indicating that the methyl group was better tolerated than the methoxy group. Finally, 8-methyl quinoline derivative **4d** (R_1_ = R_3_ = H, R_2_ = CH_3_) was found to be at least twofold less potent than that of the 6-methyl derivative **4b** (R_1_ = R_2_ = H, R_3_ = CH_3_), demonstrating that the 6-position on the quinoline moiety was more tolerated than the 8-position.

#### 2.2.2. EGFR Inhibitory Assay

Compounds **4a**, **4b**, and **7d**, the most potent derivatives as antiproliferative agents, were tested for their efficiency as EGFR inhibitors [46,47] to understand how these substances affected the EGFR enzyme. According to Table 3, compounds **4a**, **4b**, and **7d** significantly inhibited the activity of the EGFR enzyme, with IC_50_ values ranging from 64 nM to 97 nM. Compound **4a**, the most effective antiproliferative of all synthetic derivatives, showed higher potency than the standard drug erlotinib, with an IC_50_ of 64 nM as opposed to erlotinib’s IC_50_ of 70 nM. Compounds **4b** and **7d** significantly inhibited EGFR, with IC_50_ values of 93 and 97 nM, respectively, which were roughly 1.3-fold less effective than erlotinib. The outcomes of this assay supplemented cancer cell-based assay results, suggesting that EGFR-TK may be a viable target for these drugs’ antiproliferative effects.

#### 2.2.3. Apoptosis Assays

##### Effect of Compounds **4a**, **4b**, and **7d** on Caspases Cascade

The effects of derivatives **4a**, **4b**, and **7d** on caspase-3 were studied using human epithelial cancer cell line (A-549) and compared to the reference drug doxorubicin [39,48]. The results showed that **4a**, **4b**, and **7d** increased the level of active caspase-3 by 7–9 times when compared to control untreated cells, and that **4a**, **4b**, and **7d** had remarkable overexpression of caspase-3 protein level (587.50 ± 4.50, 535.50 ± 4.50, and 485.50 ± 4.25 pg/mL, respectively) when compared to the reference doxorubicin (503.2 ± 4.50 pg/mL), as shown in Table 4. Compared with control untreated cells, the most active derivatives **4a** and **4b** increased the level of active caspase-3 by 9 and 8 times, respectively, and activated caspase-3 higher than doxorubicin, Table 4.

The impact of compounds **4a** and **4b** on caspase-8 and caspase-9 was also assessed to clarify how compounds **4a** and **4b** induce apoptosis by activating the intrinsic or extrinsic route. The results showed that compound **4a** increased caspase-8 and 9 levels by 6 and 19 times, respectively, while compound **4b** showed a 4- and 18-fold increase in levels, respectively, compared to control cells. This indicates that both the intrinsic and extrinsic pathways were activated, with an effect that was more noticeable on the intrinsic pathway, because caspase-9 levels were higher, as shown in Table 4.

##### Effect of Compounds **4a**, **4b**, and **7d** on Cytochrome C Level

The concentration of Cytochrome c in a cell is crucial for activating caspases and starting the intrinsic apoptosis process [49]. The evaluation of hybrids **4a** and **4b** as inducers of Cytochrome c is summarized in Table 4. In the A-549 epithelial cancer cell line, hybrids **4a** and **4b** result in a 16- and 14-fold overexpression of Cytochrome c compared to the control. Accordingly, the results presented above show that Cytochrome c overexpression and the activation of the intrinsic apoptotic pathway by the investigated hybrids may be responsible for apoptosis.

##### Effect of Compounds **4a**, **4b**, and **7d** on BaX and Bcl2 Levels

The most effective hybrids **4a** and **4b** were further investigated for their impact on Bax and Bacl-2 levels against the A-549 epithelial cancer cell line, as shown in Table 5 [50]. The findings demonstrated that, compared to doxorubicin, **4a** and **4b** evoked a notable increase in Bax level. Compound **4a** induction of Bax (298 pg/mL) was comparable to doxorubicin (276 pg/mL), 36 times higher than control untreated A-549 cancer cells, followed by compound **4b** (284 pg/mL and 34-fold change). Finally, compound **4a** (1.05 ng/mL), compound **4b** (1.17 ng/mL), and doxorubicin (1.98 ng/mL) all reduced the level of the anti-apoptotic Bcl-2 protein in the A-549 cell line.

### 2.3. Molecular Docking Simulations 

The EGFR is a recognized receptor that binds to the EGF outside of the cell membrane and is activated, leading to the receptor’s phosphorylation. Cell survival, proliferation, and metabolism are mediated by phosphorylated EGFR. Dysfunction of the EGFR promotes uncontrolled cell development, which results in cell overgrowth and, ultimately, oncogenesis [51]. As a result, EGFR has been considered as a potential target for cancer therapy. Molecular docking analyses were carried out using the Glide software to better understand the interactions of promising compounds (**4a**, **4b**, and **7d**) with the EGFR target protein. In this methodology, Glide docking score, emodel, and Molecular mechanics with generalized Born and surface area solvation (MMGBSA) binding free energy (ΔG Bind) were kept as support for the present work. ΔG Bind is a popular method to calculate the free energy of the binding of ligands to proteins. The minimal docking score and ΔG Bind needed for complex formation between ligand and protein show good binding affinity. More negative values suggest that the ligand is buried in the receptor cavity. The mean docking score for all three compounds is −6.98 kcal/mol, and the ΔG Bind is −67 kcal/mol, as shown in Table 6. 

Compound **4a**, the most active EGFR inhibitor among the investigated compounds, also had the most significant docking score of −7.20 kcal/mol and a ΔG Bind of −75.62 kcal/mol when contrasted to Erlotinib (−9.07 and −84.86 kcal/mol, respectively). The binding interaction showed that compound **4a** formed one hydrogen bond with Arg817 at a 2.21 A° bond length, while the central naphthyl ring formed a π-cation interaction with Lys721(4.08 A°) in the EGFR kinase domain (Figure 7A). The quinolin-2(1*H*)-one and triazole portions of compound **4a** were shown to have substantial van der Waals contacts with Gly772 (−3.09 kcal/mol), Val702 (−3.95 kcal/mol), and Leu694 (−4.14 kcal/mol), which demonstrated that the molecule is entrenched within the active site.

On the other hand, Compounds **4b** and **7d** demonstrated hydrogen and hydrophobic interactions with Lys721, Met769, Phe771, and Lys405 residues in the EGFR kinase domain that were comparable to erlotinib, supporting its inhibitory activity towards EGFR.

## 3. Conclusions

In conclusion, a straightforward way to tether different quinolinones derivatives to 1,5-dinaphthol and 1,8-dinaphthol via 1,2,3-triazole linkers was elaborated based on Cu-catalyzed [3 + 2] cycloaddition of *o*-propargyl units. In the course of the introduction of the propargyl groups, an interesting dependence of the regioselectivity on the substituent found at the OH groups of the naphthalene rings was observed as Syn and Anti isomers-like. By sequential [3 + 2] cycloadditions, it was possible to link two quinolinone moieties to the naphthalene skeleton. Remarkably, this cycloaddition occurred, and all products obtained were hitherto unknown. Pharmacological screening of novel products showed interesting and promising results as EGFR inhibitors with potential apoptotic antiproliferative action.

## 4. Experimental Section

### 4.1. Chemistry 

General Details: See Appendix A.

#### 4.1.1. Starting Materials

The materials 4-Azido-2-quinoline-(1*H*)-ones **3a**–**e**, naphthalene-1,8-diol (**1**) and naphthalene-1,5-diol (**5**) (Aldrich) were used as received. The 1,8-Bis(prop-2-yn-1-yloxy)naphthalene (**2**) and 1,5-bis(prop-2-yn-1-yloxy)naphthalene (**6**) were synthesized according to the literature.

#### 4.1.2. General Procedure for the Synthesis of Compounds **4a**–**e** and **7a**–**e**

A mixture of 1,8-bis(prop-2-yn-1-yloxy)naphthalene (**2**) or 1,5-bis(prop-2-yn-1-yloxy)-naphthalene (**6**) (1.1 mmol) in 20 mL DMF, CuSO_4_.5H_2_O (0.4 mmol) and (0.4 mmol) of sodium ascorbate was stirred for 10 min at room temperature. To the above mixture, 4-azido compounds **3a**–**e** (1.0 mmol) in 20 mL DMF were added dropwise. The reaction mixture was stirred at 50 °C for 24 h. After 14 hr, another portion of sodium ascorbate (0.4 mmol) was added to the reaction mixture to prevent the reversible process for Cu (+1). The reaction mixtures were monitored with TLC. After completion, the mixture was diluted with 100 gm ice, and the formed precipitate was filtered off and washed four times with cold water to give compounds **4a**–**e** and **7a**–**e** in excellent yields.

1,8-Bis(prop-2-yn-1-yloxy)naphthalene (2) m.p 190–192 °C [31]. 

**[4,4′-(((naphthalene-1,8-diylbis(oxy))bis(methylene))bis(1*H*-1,2,3-triazole-4,1-diyl))-bis-(quinolin-2(1*H*)-one)] (4a)**. This compound was obtained as colorless powder, (85%), m.p. > 360 °C. ^1^H NMR (DMSO-*d_6_*): δ_H_ = 12.29 (s, 2H; NH-6), 9.00 (s, 2H; H-5), 7.85 (d, *J* = 8.3, 2H; H-4′), 7.66 (dd, *J* = 7.7, 7.0, 2H; H-12), 7.49 (d, *J* = 7.7, 2H; H-13), 7.48 (dd, *J* = 7.7, 6.8, 2H; H-3′), 7.45 (d, *J* = 8.2, 2H; H-10), 7.31 (d, *J* = 7.8, 2H; H-2′), 7.26 (dd, *J* = 7.5, 7.2, 2H; H-11), 6.91 (s, 2H; H-8), 5.50 ppm (s, 4H; H-4a), ^13^C NMR (DMSO-*d_6_*): δ_C_ = 160.99 (C-7), 153.37 (C-1′), 143.64 (C-4), 143.42 (C-9), 139.43 (C-13a), 131.88 (C-12), 126.46 (C-5), 126.03 (C-4a’), 125.51 (C-3′), 124.04 (C-13), 122.59 (C-11), 117.76 (C-8), 115.92 (C-10), 114.48 (C-4′, 8a’, 9a), 106.80 (C-2′), 61.67 ppm (C-4a), ^15^N NMR (DMSO-*d_6_*): δ_N_ = 247.4 (N-3), 152.3 ppm (N-6), N-1 and N-2 n/o. *m/z* = 608 (M^+^, 8). Anl. Calcd. for C_34_H_24_N_8_O_4_: C, 67.10; H, 3.97; N, 18.41; Found: C, 67.19; H, 4.11; N, 18.55.**[4,4′-(((naphthalene-1,8-diylbis(oxy))bis(methylene))bis(1*H*-1,2,3-triazole-4,1-diyl))-bis(6-methylquinolin-2(1*H*)-one)] (4b).** This compound was obtained as colorless powder (80%), m.p > 360 °C. ^1^H NMR (DMSO-*d_6_*): δ_H_ = 12.21 (s, 2H; H-6), 8.98 (s, 2H; H-5), 7.86 (d, *J* = 8.1, 2H; H-4′), 7.48 (dd, *J* = 8.3, 8.3, 4H; H-12, 3′), 7.39 (d, *J* = 8.2, 2H; H-13), 7.31 (d, *J* = 7.7, 2H; H-2′), 7.26 (s, 2H; H-10), 6.86 (s, 2H; H-8), 5.50 (s, 4H; H-4a), 2.31 ppm (s, 6H; H-11a), ^13^C NMR (DMSO-*d_6_*): δ_C_ = 160.81 (C-7), 153.37 (C-1′), 143.44 (C-4,9), 137.53 (C-13a), 133.18 (C-12), 131.74 (C-11), 126.45 (C-5), 125.63 (C-4a’), 125.29 (C-3′), 123.20 (C-10), 117.78 (C-8), 115.91 (C-13), 114.42 (C-4′, 8a’, 9a), 106.77 (C-2′), 61.67 (C-4a), ppm 20.55 (C-11a), ^15^N NMR (DMSO-*d_6_*): δ_N_ = 247.8 (N-3), 151.8 ppm (N-6). N-1 and N-2 n/o. *m/z* = 636 (M^+^, 100). Anl. Calcd. for C_36_H_28_N_8_O_4_: C, 67.91; H, 4.43; N, 17.60; Found: C, 68.07; H, 4.52; N, 17.44.**[4,4′-(((naphthalene-1,8-diylbis(oxy))bis(methylene))bis(1*H*-1,2,3-triazole-4,1-diyl))-bis-(6-methoxyquinolin-2(1*H*)-one)] (4c)**. This compound was obtained as colorless powder (77%), m.p > 360 °C. ^1^H NMR (DMSO-*d_6_*): δ_H_ = 12.20 (s, 2H; NH-6), 9.03 (s, 2H; H-5), 7.85 (d, *J* = 8.2, 2H; H-4′), 7.45 (dd, *J* = 8.2, 6.8, 2H; H-3′), 7.44 (d, *J* = 8.2, 2H; H-13), 7.34 (d, *J* = 8.7, 2H; H-12), 7.30 (d, *J* = 7.4, 2H; H-2′), 6.94 (s, 2H; H-10), 6.90 (s, 2H; H-8), 5.51 (s, 4H; H-4a), 3.67 ppm (s, 6H; H-11a), ^13^C NMR (DMSO-*d_6_*): δ_C_ = 160.52 (C-7), 154.51 (C-11), 153.32 (C-1′), 143.55 (C-4), 143.11 (C-9), 134.09 (C-13a), 126.35 (C-5), 126.03 (C-4a’), 125.47 (C-3′), 121.07 (C-12), 118.05 (C-8), 117.44 (C-13), 114.88 (C-4′, 9a), 114.44 (C-8a’), 106.69 (C-2′), 105.49 (C-10), 61.62 (C-4a), 55.33 ppm (C-11a), ^15^N NMR (DMSO-*d_6_*): δ_N_ = 247.6 (N-1), 151.3 ppm (N-6). N-2 and N-3 n/o. *m/z* = 668 (M^+^, 41). *Anal. Calcd. for* C_36_H_28_N_8_O_6_: C, 64.66; H, 4.22; N, 16.76; Found: C, 64.59; H, 4.31; N, 16.88.**[4,4′-(((naphthalene-1,8-diylbis(oxy))bis(methylene))bis(1*H*-1,2,3-triazole-4,1-diyl))-bis-(8-methylquinolin-2(1*H*)-one)] (4d)**. This compound was obtained as colorless powder (81%), m.p > 360 °C. ^1^H NMR (DMSO-*d_6_*): δ_H_ = 11.40 (bs, 2H; NH-6), 8.97 (s, 2H; H-5), 7.85 (d, *J* = 8.1, 2H; H-4′), 7.52 (d, *J* = 6.8, 2H; H-12), 7.46 (dd, *J* = 8.0, 7.4, 2H; H-3′), 7.31 (d, *J* = 7.4, 2H; H-2′), 7.22 (d, *J* = 7.2, 2H; H-10), 7.16 (dd, *J* = 7.4, 6.9, 2H; H-11), 6.91 (s, 2H; H-8), 5.50 (s, 4H; H-4a), 2.51 ppm (s, 6H; H-13b), ^13^C NMR (DMSO-*d_6_*): δ_C_ = 161.34 (C-7), 153.38 (C-1′), 144.22 (C-4), 143.39 (C-9), 137.82 (C-13a), 133.10 (C-12), 126.63 (C-5), 126.06 (C-4a’), 125.51 (C-3′), 124.42 (C-13), 122.32 (C-11), 121.80 (C-10), 117.93 (C-8), 114.85 (C-4′), 114.52 (C-8a’, 9a), 106.85 (C-2′), 61.69 (C-4a), 17.53 ppm (C-13b), ^15^N NMR (DMSO-*d_6_*): δ_N_ = 247.4 (N-3 or N-1), 149.3 ppm (N-6). N-2 and (N-1 or N-3) n/o. *m/z* = 636 (M^+^, 31). Anl. Calcd. for C_36_H_28_N_8_O_4_: C, 67.91; H, 4.43; N, 17.60; Found: C, 68.10; H, 4.39; N, 17.49.**[4,4′-(((naphthalene-1,8-diylbis(oxy))bis(methylene))bis(1H-1,2,3-triazole-4,1-diyl))-bis-(1-methylquinolin-2(1H)-one)] (4e)**. This compound was obtained as colorless powder (79%), m.p > 360 °C. NMR (DMSO-d_6_) (See Table 1) m/z = 636 (M^+^, 12). Anl. Calcd. for C_36_H_28_N_8_O_4_: C, 67.91; H, 4.43; N, 17.60; Found: C, 67.88; H, 4.55; N, 17.79.**1,5-Bis(prop-2-yn-1-yloxy)naphthalene (6)**. This compound was obtained as colorless powder (90%), m.p 150–152 °C. ^1^H NMR (DMSO-d_6_): δ_H_ = 7.75 (d, *J* = 8.4, 2H; H-4), 7.44 (dd, *J* = 8.2, 8.0, 2H; H-3), 7.10 (d, *J* = 7.7, 2H; H-2), 5.01 (d, *J* = 2.0, 4H; -OCH_2_), 3.34 (s, 2H; H-1c), ^13^C NMR (DMSO-d_6_): δ_C_ = 152.55 (C-1), 125.97 (C-4a), 125.43 (C-3), 114.23 (C-4), 106.74 (C-2), 79.12 (C-1b), 78.38 (C-1c), 55.99 (-OCH_2_). M/z = 236 (M^+^, 35). Anl. Calcd. for C_16_H_12_O_2_: C, 81.34; H, 5.12; Found: C, 81.44; H, 4.97.**[4,4′-(((naphthalene-1,5-diylbis(oxy))bis(methylene))bis(1*H*-1,2,3-triazole-4,1-diyl))-bis-(quinolin-2(1*H*)-one)] (7a)**. This compound was obtained as colorless powder (86%), m.p > 360 °C. ^1^H NMR (DMSO-d_6_): δ_H_ = 12.21 (s, 2H; NH-6), 8.98 (s, 2H; H-5), 7.84 (d, *J* = 8.4 Hz, 2H; H-4′), 7.77 (d, *J* = 8.5 Hz, 2H; H-13), 7.51–7.26 (m, 9H; H-12,3′,10,11,2′,8), 6.86 (s, 2H; H-2′), 5.50 (s, 4H; H-4a), ^13^C NMR (DMSO-d_6_): δ_C_ = 160.84 (C-7), 153.11 (C-1′), 143.42 (C-4), 142.64 (C-9), 140.11 (C-13a), 133.18 (C-12), 126.45 (C-5), 126.07 (C-10), 125.50 (C-3′), 125.29 (C-4a’), 124.50 (C-11), 123.20 (C-9a), 117.24 (C-8), 115.91 (C-13), 114.42 (C-4′), 106.77 (C-2′), 61.88 (C-4a). *m/z* = 608 (M^+^, 20). Anl. Calcd. for C_34_H_24_N_8_O_4_: C, 67.10; H, 3.97; N, 18.41; Found: C, 66.95; H, 3.88; N, 18.58.**[4,4′-(((naphthalene-1,5-diylbis(oxy))bis(methylene))bis(1H-1,2,3-triazole-4,1-diyl))-bis-(6-methylquinolin-2(1H)-one)] (7b).** This compound was obtained as colorless powder (73%), m.p > 360 °C. ^1^H NMR (DMSO-d_6_): δ_H_ = 12.22 (s, 2H; H-6), 8.96 (s, 2H; H-5), 7.96 (d, *J* = 8.0, 2H; H-4′), 7.85 (d, *J* = 8.4, 4H; H-12, 3′), 7.51–7.26 (m, 6H; H-13,2′,10), 6.86 (s, 2H; H-8), 5.50 (s, 4H; H-4a), 2.09 ppm (s, 6H; H-11a). m/z = 636 (M^+^, 31). Anl. Calcd. for C_36_H_28_N_8_O_4_: C, 67.91; H, 4.43; N, 17.60; Found: C, 67.83; H, 4.38; N, 17.78.**[4,4′-(((naphthalene-1,5-diylbis(oxy))bis(methylene))bis(1H-1,2,3-triazole-4,1-diyl))-bis-(6-methoxyquinolin-2(1H)-one)] (7c).** This compound was obtained as colorless powder (78%), m.p > 360 °C. ^1^H NMR (DMSO-d_6_): δ_H_ = 12.19 (s, 2H; NH-6), 9.02 (s, 2H; H-5), 7.84 (d, *J* = 6.2, 2H; H-4′), 7.45 (d, *J* = 6.0, 2H; H-13), 7.31 (dd, *J* = 9.1, 8.2, 2H; H-12), 6.94 (d, *J* = 8.0, 2H; H-3′), 7.30 (d, *J* = 7.4, 2H; H-10), 6.94 (s, 2H; H-11), 6.70 (s, 2H; H-8), 5.51 (s, 4H; H-4a), 3.32 ppm (s, 6H; H-11a), ^13^C NMR (DMSO-d_6_): δ_C_ = 160.52 (C-7), 154.51 (C-11), 153.32 (C-1′), 143.55 (C-4), 143.11 (C-9), 134.09 (C-13a), 126.35 (C-5), 126.03 (C-4a’), 125.47 (C-3′), 121.07 (C-12), 118.05 (C-8), 117.44 (C-13), 114.88 (C-4′, 9a), 114.44 (C-8a’), 106.69 (C-2′), 105.49 (C-10), 61.62 (C-4a), 55.33 ppm (C-11a), ^15^N NMR (DMSO-d_6_): δ_N_ = 151.4 ppm (N-6). N-1, N-2 and N-3 n/o. m/z = 668 (M^+^, 8). Anl. Calcd. for C_36_H_28_N_8_O_6_: C, 64.66; H, 4.22; N, 16.76; Found: C, 64.79; H, 4.18; N, 16.59.**[4,4′-(((naphthalene-1,5-diylbis(oxy))bis(methylene))bis(1*H*-1,2,3-triazole-4,1-diyl))-bis-(8-methylquinolin-2(1*H*)-one)] (7d).** This compound was obtained as colorless powder (88%), m.p > 360 °C. ^1^H NMR (DMSO-*d_6_*): δ_H_ =11.38 (bs, 2H; NH-6), 8.96 (s, 2H; H-5), 7.96–690 (m, 12H; H-4′,12,3′,2′,10,8), 5.50 (s, 4H; H-4a), 2.50 ppm (s, 6H; H-13b). *m/z* = 636 (M^+^, 23). Anl. Calcd. for C_36_H_28_N_8_O_4_: C, 67.91; H, 4.43; N, 17.60; Found: C, 68.06; H, 4.52; N, 17.79.**[(4,4′-(((naphthalene-1,5-diylbis(oxy))bis(methylene))bis(1H-1,2,3-triazole-4,1-diyl))-bis-(1-methylquinolin-2(1H)-one)] (7e)**. This compound was obtained as colorless powder (74%), m.p > 360 °C. ^1^H NMR (DMSO-d_6_) (See Table 1). m/z = 636 (M^+^, 58). Anl. Calcd. for C_36_H_28_N_8_O_4_: C, 67.91; H, 4.43; N, 17.60; Found: C, 67.78; H, 4.39; N, 17.74.

### 4.2. Biology

Appendix A contains information on all biological experimental tests.

### 4.3. Docking Study

Molecular docking simulations were performed using MOE^®^ software within EGFR protein crystal structure with erlotinib as a co-crystallized ligand (PDB ID: 1M17). Docking protocol and other experimental details were used exactly as reported elsewhere [52,53,54]. See Appendix A.

## Data Availability

The study did not report any data.

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
