# Peer review of "Design, Synthesis and Biological Evaluation of Syn and Anti-like Double Warhead Quinolinones Bearing Dihydroxy Naphthalene Moiety as Epidermal Growth Factor Receptor Inhibitors with Potential Apoptotic Antiproliferative Action"

_molecules, 2022, doi:10.3390/molecules27248765_

Round 1
Reviewer 1 Report
This manuscript is well written and structured. However, I would invite the authors to consider the following comments to improve its overall quality.
Page 1 line 30: Please introduce or briefly describe compounds 4a-e and 7a-e before using them in the abstract and/or text. The same applies for compounds 4a, 4b and 7d. Likewise, please define acronyms such as EGFR and GI50 before using them.
Line 67: Please correct the typo at the beginning of the sentence starting at the end of the line.
Why does the manuscript not include a section for the materials and methods? Does the experimental section substitute this?
Line 250: the scheme of compounds 4a-e and 7a-e should be labeled independently of Table 2.
Author Response
Reviewer #1
This manuscript is well written and structured. However, I would invite the authors to consider the following comments to improve its overall quality.
- Page 1 line 30: Please introduce or briefly describe compounds 4a-e and 7a-e before using them in the abstract and/or text. The same applies for compounds 4a, 4b and 7d.
Response
Done as advised
- Likewise, please define acronyms such as EGFR and GI50 before using them.
Response
Done as advised
- Line 67: Please correct the typo at the beginning of the sentence starting at the end of the line.
Response
Done as advised
- Why does the manuscript not include a section for the materials and methods? Does the experimental section substitute this?
Response
Yes sir, please refer to sections 4.1, 4.2, and 4.3
- Line 250: the scheme of compounds 4a-e and 7a-e should be labeled independently of Table 2.
Response
I appreciate your thoughtful comment. For the readers' visibility, we have only included the structures of these compounds—not a scheme.
Reviewer 2 Report
This manuscript by El-Sheref et al. describes “Design, synthesis, and biological evaluation of syn and anti-like double warhead quinolinones bearing dihydroxy naphthalene moiety as EGFR inhibitors with potential apoptotic antiproliferative action”. These compounds displayed potent cytotoxicity against cancer cell lines and EGFR inhibitory activity. I would recommend this to be published after comments as below
1. Authors should perform annexin binding assay to find out percentage of cancer cells undergoing apoptosis.
2. Which cancer cells were used to perform caspase studies in section 2.2.3.1. Please mention this in the text.
3. Briefly explain why A549 cells were used to apoptotic assays.
4. Figure 7 is not clear. Authors should improve the visibility of amino acids and interactions in this figure.
5. Typing mistakes like deversity (line 98) should be corrected. The whole manuscript can be checked for typing mistakes.
Author Response
Reviewer #2
This manuscript by El-Sheref et al. describes “Design, synthesis, and biological evaluation of syn and anti-like double warhead quinolinones bearing dihydroxy naphthalene moiety as EGFR inhibitors with potential apoptotic antiproliferative action”. These compounds displayed potent cytotoxicity against cancer cell lines and EGFR inhibitory activity. I would recommend this to be published after comments as below
- Authors should perform annexin binding assay to find out percentage of cancer cells undergoing apoptosis.
Response
We appreciate the reviewer's feedback; however, these compounds are currently being investigated for further biological testing and will be published as future work.
- Which cancer cells were used to perform caspase studies in section 2.2.3.1. Please mention this in the text.
Response
Done as advised
- Briefly explain why A549 cells were used to apoptotic assays.
Response
The tested compounds had the greatest potency against A549
- Figure 7 is not clear. Authors should improve the visibility of amino acids and interactions in this figure.
Response
Done as advised, please see the attachment
- Typing mistakes like deversity (line 98) should be corrected. The whole manuscript can be checked for typing mistakes.
Response
Done as advised
Round 2
Reviewer 2 Report
Authors responded to comments and corrected manuscript at required places. I would recommend to accept it.
Page 10, line 224; page 11, line 275; A-549 cancer cell line is human lung carcinoma epithelial cells. This can be modified.